# Upcycling Text-to-Image Diffusion Models for Multi-Task Capabilities

**Ruchika Chavhan** [1] **Abhinav Mehrotra** [1] **Malcolm Chadwick** [1] **Alberto Gil Couto Pimentel Ramos** [1]
**Luca Morreale** [1] **Mehdi Noroozi** [1] **Sourav Bhattacharya** [1]

## Abstract

Text-to-image synthesis has witnessed remarkable advancements in recent years. Many attempts have been made to adopt text-to-image models to support multiple tasks. However, existing approaches typically require resource-intensive retraining or additional parameters to accommodate for the new tasks, which makes the model inefficient for on-device deployment. We propose *Multi-Task Upcycling* (MTU), a simple yet effective recipe that extends the capabilities of a pretrained text-to-image diffusion model to support a variety of image-to-image generation tasks. MTU replaces Feed-Forward Network (FFN) layers in the diffusion model with smaller FFNs, referred to as *experts*, and combines them with a dynamic routing mechanism. To the best of our knowledge, MTU is the first multi-task diffusion modeling approach that seamlessly blends multi-tasking with on-device compatibility, by mitigating the issue of parameter inflation. We show that the performance of MTU is on par with the single-task fine-tuned diffusion models across several tasks including *image editing, super-resolution*, and *inpainting*, while maintaining similar latency and computational load (GFLOPs) as the single-task fine-tuned models.

## 1. Introduction

Text-to-image (T2I) generation with diffusion models is rapidly gaining traction across diverse applications, with foundational models such as DALLE2 (Ramesh et al., 2022), MidJourney, Stable Diffusion (Rombach et al., 2022; Zhang et al., 2023; Podell et al., 2023; Stability AI, 2023; Lin et al., 2024), and Diffusion Transformers (DiT) (Peebles & Xie, 2022; Gao et al., 2024; Zhuo et al., 2024; Xie et al., 2024;

---

[1]Samsung AI Center, Cambridge. Correspondence to: Ruchika Chavhan <r2.chavan@samsung.com>.

*Proceedings of the 42$^{nd}$ International Conference on Machine Learning*, Vancouver, Canada. PMLR 267, 2025. Copyright 2025 by the author(s).

Esser et al., 2024) at the forefront. Thanks to the opensourcing efforts, developers have the opportunity to finetune them for a variety of creative use-cases. The growing demand for generative AI applications has also contributed to the requirement of deploying the *state-of-the-art* (SOTA) models on personal and edge devices to address data-privacy and the cost of cloud hosting.

Great effort has been made to optimize these foundation models for edge deployment by making them smaller, faster, and resource-efficient (Zhao et al., 2025; Castells et al., 2024; Zhang et al., 2024b). These optimization strategies include distillation of models to reduce size (Xiang et al., 2024; Tang et al., 2023; Fang et al., 2023), reduction of the frequency of model calls (Salimans & Ho, 2022; Meng et al., 2023; Kang et al., 2025; Zhu et al., 2025; Yin et al., 2024; Noroozi et al., 2024), reduction of on-device memory and latency requirements through quantization (Li et al., 2023; He et al., 2023; Wang et al., 2024), and removal of computationally intensive operations (Zhao et al., 2023).

As SOTA diffusion models have demonstrated capabilities to support various use cases through fine-tuning, there is also a growing interest to develop a single model to perform multiple image-to-image (I2I) tasks, like image editing (Brooks et al., 2023), super-resolution (Moser et al., 2024), in/out-painting (Corneanu et al., 2024; Wasserman et al., 2024). However, incorporating multiple tasks presents significant challenges. Some approaches adopt universal modeling (Ye & Xu, 2024; Zhang et al., 2024a; Bao et al., 2023) to learn a joint probability distribution that unifies multiple modalities within a common diffusion space. Other approaches rely on designing models with specialized components tailored to specific modalities or tasks (Xu et al., 2023; Tang et al., 2024). This allows the diffusion space to vary while partitioning the computational graph based on the task, offering modularity and flexibility. However, these methods significantly increase the model size and parameter count, making them computationally inefficient. Moreover, these approaches face scalability issue as the computational requirements grow significantly with the increase in the number of tasks and modalities. Thus, a significant gap remains in efficiently adapting diffusion models to multiple tasks, while ensuring they are suitable for on-device deployment.

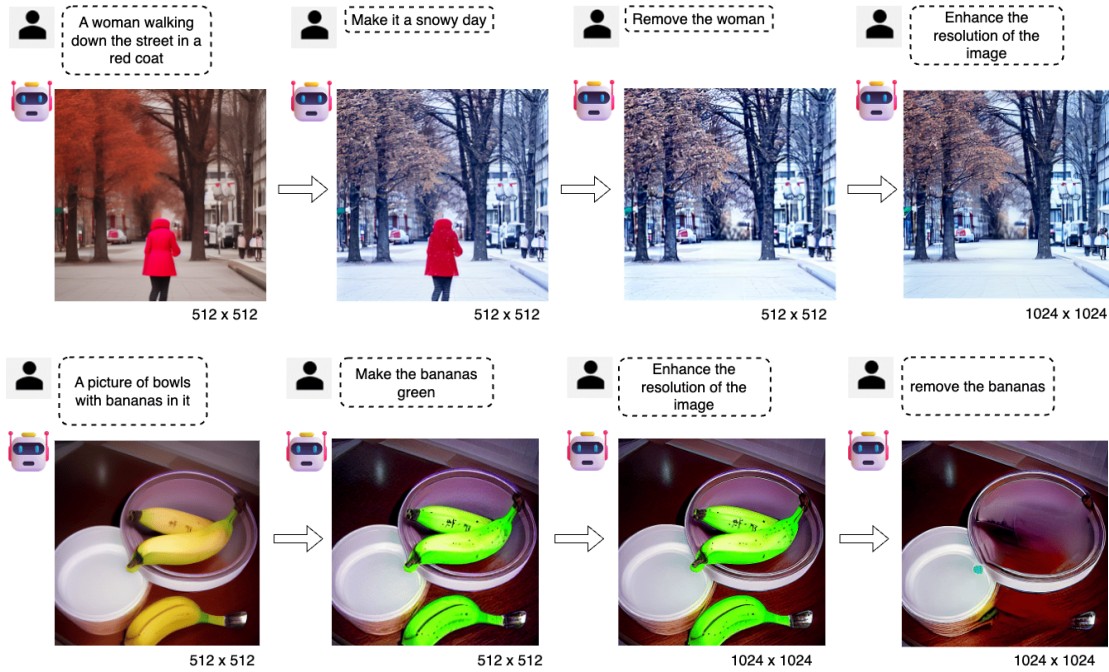

*Figure 1.* A chatbot showcasing a potential use case of Multi-Task Upcycling. Our approach efficiently upcycles pre-trained text-to-image models, enabling them to perform multiple image generation tasks using a single backbone.

To bridge the gap, we introduce the concept of *Multi-Task Upcycling* (MTU) of T2I diffusion models. MTU transforms single-task T2I models into image-generation generalists, i.e., a single model is capable of handling multiple tasks. The concept of *upcycling* is widely used in the field of Large Language Models (LLMs) to transform dense pre-trained models into sparse Mixture-of-Experts (MoE) (He et al., 2024; Komatsuzaki et al., 2023; Jiang et al., 2025), but it has not yet been explored for diffusion models. Unlike previous multi-task diffusion models, our approach avoids the need to significantly increase model parameters to support additional new tasks. Instead, it only requires retraining a few components with new multi-task data, while ensuring that the model remains both efficient and scalable for multi-task learning. MTU is particularly effective in the following two main scenarios: (i) extending an on-device T2I diffusion model to support multiple tasks without increasing the computational requirement, (ii) obtaining a multi-task diffusion model with an existing T2I model acting as a strong prior. We demonstrate a potential use case of our proposed approach as a *chatbot*, illustrated in Figure 1, where a single model seamlessly handles multiple tasks such as image editing, inpainting, and super-resolution based on user requests.

Our method is inspired by the *empirical* observation that when a T2I model is fine-tuned for a new task, the parameters in the Feed-Forward Network (FFN) layers undergo a significant shift, compared to the other layers of a diffusion model. Building on this insight and inspired by upcycling literature, we convert a single FFN layer in pre-trained models into a number of smaller FFN *experts*. These experts are fine-tuned simultaneously on multiple tasks while keeping the rest of the model frozen. A router mechanism dynamically learns to combine the outputs of these individual experts, guided by task-specific embeddings. We evaluate our method on latent diffusion models, such as *SDv1.5* and *SDXL*, across various image-to-image tasks, including image editing, super-resolution, and inpainting. Our results demonstrate that we can develop models that are *iso-FLOP*, i.e., having the same FLOPs as their pre-trained counterparts, while achieving performance comparable to single-task models. Our upcycled SDXL achieves a FID of 3.9 on T2I generation, while maintaining the same computational cost of 1.54 TFLOPs as the pre-trained model.

## 2. Related Work

**Multi-task Diffusion Models:** There are two primary approaches in this area. The first is universal modeling (Ye & Xu, 2024; Zhang et al., 2024a; Bao et al., 2023; Chen et al., 2024), which aims to learn a joint probability distribution unifying multiple modalities within a shared diffusion space. For instance, instead of learning $p(\text{image}|\text{text})$ as in single-task diffusion models, it learns $p(\text{text}, \text{image})$, the joint probability distribution for text and image. Text-to-image generation can then be performed by marginalizing the joint distribution. However, a key drawback of this

approach is its scalability as the entire model needs to be re-trained when new tasks are added. The second approach involves designing models with specialized components tailored to specific modalities or tasks (Xu et al., 2023; Tang et al., 2024). This enables the diffusion space to vary across tasks, while partitioning the computational graph for modularity and flexibility. Despite these advantages, this method significantly increases model size and parameter count, reducing efficiency. For both approaches, computational requirements scale rapidly with the number of tasks and modalities, resulting in scalability challenges.

**Sparse Upcycling of Pre-trained Models:** Upcycling has been explored in various studies as a method for transforming trained dense models into Mixture-of-Experts (MoE) frameworks (He et al., 2024; Komatsuzaki et al., 2023; Jiang et al., 2025). The concept of upcycling arose from the challenges of training sparse MoE models from scratch, as such training is highly unstable and sensitive to hyperparameters. Upcycling offers a solution by starting with a pre-trained dense model, which is often readily available online, and transforming it into an MoE model to enhance performance and capacity. At the core of any upcycling method, discussed in (He et al., 2024; Komatsuzaki et al., 2023; Jiang et al., 2025), lies expert architecture design, initialization techniques, and routing strategies within the MoE layer.

The Feed Forward Networks (FFNs) within pre-trained LLMs, which are two-layer MLPs with hidden dimension $d_{\text{ffn}}$, are replaced by MoE layers. An MoE layer comprises $N$ FFN experts denoted by $\{E_1, E_2, \cdots, E_N\}$ and a router that learns to assign tokens in the input to appropriate experts. Let us denote the hidden dimension of experts as $d_{\text{expert}}$. In sparse upcycling methods, the parameters of these experts are initialized using those of the pre-trained model. Various routing mechanisms have been utilized in different works (Lewis et al., 2021; Clark et al., 2022), with the Expert Choice router (Zhou et al., 2022) and standard topK routing (Shazeer et al., 2017) being the most commonly used methods for computing the output of the MoE layer.

Many studies have proposed that increasing the number of experts a token is routed to, while simultaneously reducing the dimensions of each expert such that $d_{\text{expert}} < d_{\text{FFN}}$, can be a more efficient approach. This model is referred to as a *fine-grained MoE* model (Krajewski et al., 2024) and the ratio $G = d_{\text{FFN}}/d_{\text{expert}}$ is termed as *granularity*. Reducing the dimensions of the experts decreases the FLOPs per expert, which in turn permits an increase in the topK (the number of experts a token is routed to) proportional to the reduction in expert size, all while maintaining the overall FLOPs count.

We draw inspiration from the concept of *fine-grained MoE* models in the LLM literature and adapt it for multi-tasking in diffusion models. In the following sections, we provide the motivation for focussing on the FFN blocks of diffusion

models, followed by a detailed outline of our multi-task upcycling approach.

## 3. Preliminaries

**Latent Diffusion Models:** Latent Diffusion Models (LDMs) (Rombach et al., 2022; Zhang et al., 2023) are based on Diffusion Models (DMs) (Ho et al., 2020; Song et al., 2021), that learn to reverse a forward Markov process in which noise is incrementally added to input images over multiple time steps $t \in [0, T]$. We denote an RGB image by $\mathbf{x}_0 \in \mathbb{R}^{3 \times H \times W}$, where $H$ and $W$ correspond to the height and width of the image. An encoder $\mathcal{E}$ transforms the input image $\mathbf{x}_0$ into a latent representation $\mathbf{z}_0 \in \mathbb{R}^{c \times h \times w}$, where $h$ and $w$ represent the height and width of the downscaled encoded image, and $c$ indicates the number of channels in latent space. During training, a noisy latent $\mathbf{z}_t$ at time $t$ is obtained from a real image's latent $\mathbf{z}_0$ by $\mathbf{z}_t = \sqrt{a_t}\mathbf{z}_0 + \sqrt{1 - a_t}\epsilon$, where $\epsilon \sim \mathcal{N}(\mathbf{0}, \mathbf{I})$ and $a_t$ is a parameter that gradually decays over time. A denoiser $f_\theta(.)$ is then trained to predict the noise added to $\mathbf{z}_t$ conditioned on the input text embedding $\mathbf{c}_T$. This enables the reconstruction of $\mathbf{z}_0$ by subtracting the predicted noise from $\mathbf{z}_t$. To achieve this, the denoiser is trained to predict the noise by stochastically minimizing the objective: $\mathcal{L}(\mathbf{z}, \mathbf{c}_T) = \mathbb{E}_{\epsilon, \mathbf{x}, \mathbf{c}_T, t}\left[\|\epsilon - f_\theta(\mathbf{z}_t, \mathbf{c}_T, t)\|\right]$. A decoder $\mathcal{D}$, then maps the denoised $\hat{\mathbf{z}_0}$ back to the pixel space. During inference, given a text prompt $\mathbf{c}_T$, a noisy latent embedding $\mathbf{z}_T$ is sampled and iteratively denoised over $T$ steps to produce $\hat{\mathbf{z}_0}$, which is decoded into the final image. Typically, the encoder and decoder are derived from a pre-trained autoencoder that remains frozen during training.

**Fine-tuning LDMs for Image-to-Image Generation Tasks:** The objective of image-to-image (I2I) generation tasks is to transform an input image $\mathbf{c}_I$ into a target image $\mathbf{c}_{\text{target}}$ based on an edit prompt $\mathbf{c}_T$. In I2I literature (Brooks et al., 2022), $\mathbf{c}_T$ and $\mathbf{c}_I$ are commonly referred to as the text and image *conditions*, respectively. The target image and input image are encoded by an encoder $\mathcal{E}$ to obtain the latent representations $\mathbf{z}_{\text{target}}$ and $\mathbf{z}_c$, respectively. To train a diffusion model, noise is added to $\mathbf{z}_{\text{target}}$ to obtain $\mathbf{z}_t$ by $\sqrt{a_t}\mathbf{z}_{\text{target}} + \sqrt{1 - a_t}\epsilon$, where $\epsilon \sim \mathcal{N}(\mathbf{0}, \mathbf{I})$ and $a_t$ is a parameter that is scheduled similar to T2I diffusion models. A denoiser $f_\theta$ is then trained to predict the noise added to a noisy input latent $\mathbf{z}_{\text{target}}$, given the image condition $\mathbf{z}_c$ and the text instruction $\mathbf{c}_T$. To achieve this, the image condition $\mathbf{z}_c$ is concatenated with the noisy latents $\mathbf{z}_t$, and the resulting tensor is provided as input to the denoiser. To accommodate the additional channels introduced by the image conditioning, the first convolutional layer is modified to include extra input channels, while the rest of the architecture remains unchanged. The training process involves minimizing the following latent diffusion objective:

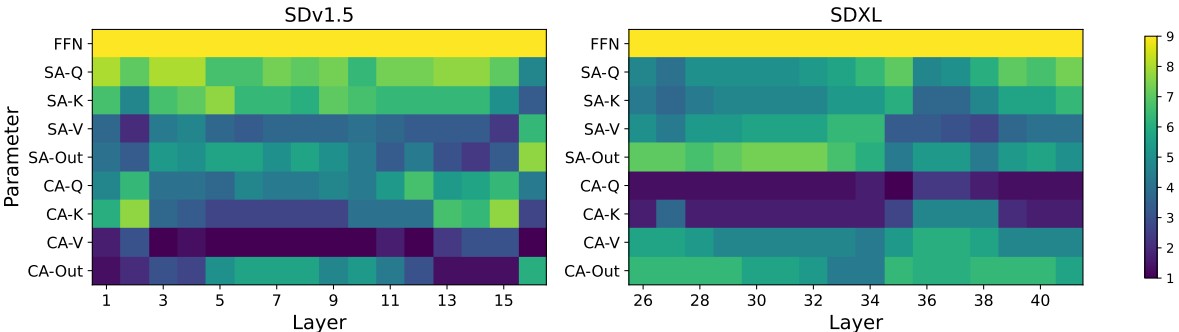

*Figure 2.* We analyze the deviation between fine-tuned weights $\theta_f^\tau$ and pre-trained initialization $\theta_p$ across different layers in the LDM (i.e., $\Phi^\tau = ||\theta_f^\tau - \theta_p||$) and rank them accordingly. We present the average rank of these deviations across all tasks. The x-axis represents layer depth, while the y-axis indicates the component type. FFN layers show the highest deviation, suggesting they specialize in adapting to downstream tasks.

|  | Image Editing (IE) | Super Resolution (SR) | Inpainting (IP) |
|---|---|---|---|
|  | I-T Dir Sim ↑ | LPIPS ↓ | I-I Dir Sim ↑ |
| SA | 17.6 | 25.0 | 42.9 |
| CA | 16.9 | 27.5 | 40.1 |
| FFNs | **17.8** | **23.7** | **46.7** |

*Table 1.* Quantitative comparison of fine-tuning different components (SA, CA, and FFNs) of the diffusion model over image-to-image generation tasks. Fine-tuning FFNs leads to better performance on image-to-image tasks.

$\mathcal{L}(\mathbf{z}, \mathbf{c}_T, \mathbf{z}_c) = \mathbb{E}_{\epsilon, \mathbf{x}, \mathbf{c}_T, \mathbf{z}_c, t} [||\epsilon - f_\theta(\mathbf{z}_t, \mathbf{c}_T, \mathbf{z}_c, t)||]$. Pre-trained text-to-image models, such as Stable Diffusion, are commonly used as initialization to leverage their extensive text-to-image generation capabilities.

## 4. Motivation

Previous studies (Loshchilov & Hutter, 2017; Kirkpatrick et al., 2017) in the area of transfer learning and domain adaptation have shown that parameters undergoing significant changes during fine-tuning are more relevant to the specific task, while those with minimal changes are either already well-aligned with the task due to pre-training or less critical for fine-tuning. Inspired from these findings, we identify a subset of parameters within the diffusion models that show the highest deviations from the pre-trained initialization. In this experiment, we consider two Latent Diffusion Models, namely SDv1.5 and SDXL. We fine-tune them on three tasks: (i) image editing, (ii) super-resolution, and (iii) inpainting. See the appendix for further details on the dataset and training procedure in Section A.2.

Let $\theta_p$ be the weights of a pre-trained diffusion model, and $\theta_f^\tau$ the fine-tuned weights for a specific task $\tau$. The distance between the fine-tuned weights and the initialization is computed as $\Phi^\tau = ||\theta_f^\tau - \theta_p||$, where $|| \cdot ||$ is the Frobenius norm. For each task, we define $\Phi^\tau$ for all LDM components: Self-Attention ($\Phi_{SA}^\tau$), Cross-Attention ($\Phi_{CA}^\tau$), and

Feed-Forward Network ($\Phi_{FFN}^\tau$) blocks. Within the SA and CA blocks, we further consider the Q, K, V, and output layer matrices. We rank $\Phi_{SA}^\tau$, $\Phi_{CA}^\tau$, and $\Phi_{FFN}^\tau$ for each layer across all tasks and then compute the average rank across I2I tasks. Figure 2 illustrates layer-wise average ranks for I2I tasks. Higher ranks signify greater deviation from the pre-trained initialization. Our experiments show that FFN layers deviate the most from the pre-trained weights to adapt to downstream I2I tasks. This is also demonstrated in Figure 6 in the appendix, which presents measured deviation values.

To further validate this observation, we fine-tune each component of the attention block separately and compare their performance. Details on the metrics reported are in Section 6. As shown in Table 1, fine-tuning the FFN layers in diffusion models consistently yields better results compared to tuning other components. In other words, FFNs specialize in solving a downstream task, while the other parameters in the attention block learn more general features. In this paper, we build on these findings, and propose an approach for Multi-Task Upcycling for Diffusion Models.

## 5. Methodology: Multi-task Upcycling for Diffusion Models

In MTU, we segment the FFN component into experts such that their weighted combination solves specific tasks without the need of extra parameters. MTU comprises four key steps: (i) split the pre-trained model's FFNs into smaller FFN experts, (ii) design a router to dynamically combine the outputs of these experts, (iii) define task-specific input processing layers, and (iv) design the loss function to train the FFN experts and the router. Note that only the FFN experts, the router, and the task-specific input processing layers are trainable parameters. An overview of our method is presented in Figure 3.

**Expert architecture:** We replace each FFN block in the pre-

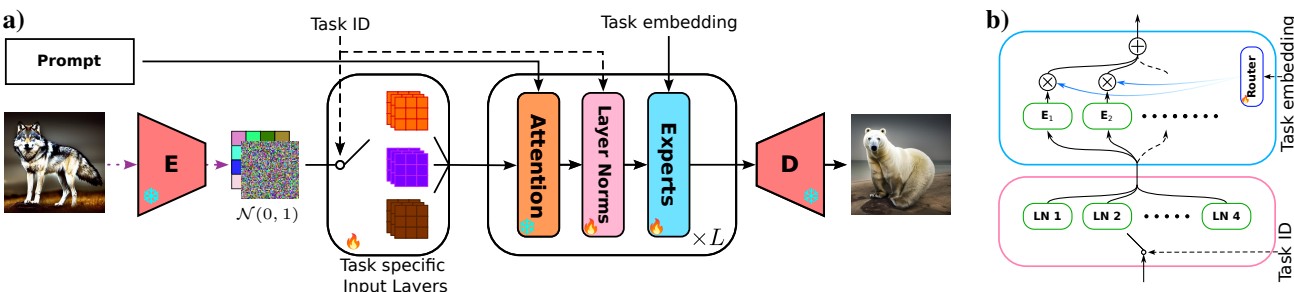

*Figure 3.* (a) Overview: We introduce Multi-task Upcycling (MTU), a method for transforming a pre-trained text-to-image model to support multiple tasks. (b) In MTU, we replace the FFN layer in the pre-trained model with a set of smaller experts, which are dynamically combined using a router mechanism.

trained denoiser architecture with $N$ experts $\{E_1^l, \cdots, E_N^l\}$, each with dimension $d_{\text{expert}}^l$, where $l$ is the layer index. To ensure that the upcycled model maintains a parameter count similar to the pre-trained model, we constrain $d_{\text{expert}}^l = d_{\text{FFN}}^l/N$, where $d_{\text{FFN}}^l$ denotes the dimension of $l$-th FFN block in the pre-trained model.

Upcycling methods typically initialize the parameters of the experts from the pre-trained model. However, in our case, since the size of the FFN in the pre-trained model differs from the upcycled model, copying the dense MLP weights to the upcycled model structure becomes non-trivial. To address this, we segment the dense layer into $N$ shards along the appropriate dimension and copy each shard into the corresponding expert. We denote the set of parameters in all the experts within the entire model as $\theta_E$.

**Router:** Let $\mathcal{T}$ denote the set of tasks. For each task $\tau \in \mathcal{T}$, we define a learnable task embedding $e_\tau \in \mathbb{R}^{d_{\text{task}}}$. We add a router to a layer $l$ as: $g(\cdot \; ; \; \theta_r^l) : \{\{e_\tau\}_{\mathcal{T}} \to \mathbb{R}^N\}$, where each expert is assigned a weight based on $e_\tau$. The output of the FFN block for an input $x_\tau$ corresponding to task $\tau$ is computed as follows:

$$w_i^l = \text{softmax}(g(e_\tau; \theta_r^l))_i$$
$$E_{\text{FFN}}^l(x_\tau) = \sum_{i=1}^{N} w_i^l \times E_i^l(x_\tau), \tag{1}$$

where $w_i^l$ represents the weight assigned to the $i$-th expert, and $E_i(x_\tau)^l$ is the output of the $i$-th expert for input $x_\tau$ in layer $l$ of the model. A key advantage of this formulation is that the task-specific weights can be pre-calculated with minimal computational overhead using the task identifier.

**Task-specific Layer Norms:** Our empirical observation suggests that incorporating a task-specific layer normalization step before each expert significantly improves model performance. The arrangement of these layer normalization steps is illustrated in Figure 3(b). Our experiments show that FFN layers exhibit significantly different distributions

across tasks, and thus the task-specific layer normalization facilitates the learning of these distinct distributions in the upcycled model. We denote the set of all task-specific convolution layers as $\Psi_L$.

**Task-specific input layers:** As outlined in Section 3, single-task image-to-image models share the same architecture, differing only in the input convolution layer, which accommodates additional channels from image conditioning. In our upcycled architecture, we introduce separate task-specific input convolution layers, denoted by $\psi_\tau(\cdot)$, to handle the varying conditioning distributions for different tasks. We denote all the set of task-specific convolution layers as $\Psi_C$.

**Multi-task Loss:** For a task $\tau$, let $z_0^\tau$ represent the encoded images, $z_t^\tau$ the noisy latents at time step $t$. We feed $z_t^\tau$ and the text prompt $c_T^\tau$ to the denoiser $f$, and task it to predict the noise $\epsilon$. Note that in the case of image conditioning, $z_t^\tau$ is the concatenation of the noisy latents and the encoded context image $c_I^\tau$ along the channel dimension.

To reduce the computational burden, we pre-compute the expert weights using the task identifiers $e_\tau$. We use these weights to combine the expert outputs as shown in Equation 1. While the shared parameters of the model are kept frozen, the task-specific layer norms $\Psi_L$ and input layers $\Psi_C$, experts $\theta_E$ and routers $\theta_R$ are trained to optimize a multi-task objective with the loss $L$ defined as:

$$L = \sum_{\tau \in \mathcal{T}} \mathbb{E}_{\epsilon, \mathbf{x}^\tau, \mathbf{c}_T^\tau, \mathbf{c}_I^\tau, t} \|\epsilon - f_\theta\left(\Psi_C(\mathbf{z}_t^\tau), \mathbf{c}_T^\tau, \mathcal{E}(\mathbf{c}_I^\tau), t\right)\| \tag{2}$$

## 6. Experimental Settings

**Model Architectures:** We evaluate our method on two Stable Diffusion-based models—SDv1.5 (Rombach et al., 2022) and SDXL (Podell et al., 2023) consisting of 860M and 2.6B parameters in the denoiser component respectively. In both models, the denoiser is a UNet composed of transformer blocks with Self-Attention (SA), Cross-Attention (CA), Feed-Forward Networks (FFNs), and residual blocks.

| | Multi-task | Model | TFLOPs | Parameters | Text-to-Image (T2I) | Image Editing (IE) | Super Resolution (SR) | Inpainting (IP) |
|---|---|---|---|---|---|---|---|---|
| | | | | | FID ↓ | I-T Direction Similarity ↑ | LPIPS ↓ | I-I Directional Similarity ↑ |
| SD v1.5 | × | T2I (Rombach et al., 2022)
IE (Brooks et al., 2023)
SR (Rombach et al., 2022)
IP (Yildirim et al., 2023) | 0.67 | 860M | 12.9
–
–
– | –
15.4
–
– | –
–
38.0
– | –
–
–
**46.5** |
| | ✓ | VD (Xu et al., 2023)
Unidiffuser (Bao et al., 2023) | 0.87
0.83 | 1.1B
952M | 10.1
7.4 | 14.2
– | –
– | –
– |
| | ✓ | MTU (Ours) | 0.68 | 869M | **7.2** | **17.2** | 24.8 | 44.0 |
| SDXL | × | T2I (Podell et al., 2023)
IE (Brooks et al., 2023)
SR
IP | 1.53 | 2.6B | 4.1
–
–
– | –
17.3
–
– | –
–
26.9
– | –
–
–
43.2 |
| | ✓ | MTU (Ours) | 1.54 | 2.6B | **3.9** | **20.1** | 26.5 | **44.2** |

*Table 2.* Quantitative comparison of Multi-task Upcycling (MTU) against single-task and multi-task baselines. We consider $N = 4$ for SDv1.5 and $N = 1$ for SDXL. MTU consistently surpasses baselines while preserving computational efficiency.

For multi-task upcycling, we consider all transformer blocks in SDv1.5 (16 blocks) and SDXL (70 blocks).

Each router network $g(\cdot \mid \theta_r^l)$ is implemented as a two-layer MLP with ReLU activation. As shown in Equation 1, we then apply a softmax function over the router's predictions to obtain the weights assigned to each expert.

**Downstream Image Synthesis Tasks:** We consider four tasks in our study, including Text-to-Image (T2I) generation, Image Editing (IE), Super Resolution (SR), and Image Inpainting (IP). Since the MTU model is initialized with a pre-trained T2I model, we include T2I as one of the tasks to ensure the multi-task model maintains its text-to-image generation capability.

**Datasets:** Since each task requires different data configurations, we use the following datasets to train the MTU model.

- T2I: We use the COCO Captions dataset (Lin et al., 2015), a large collection of image-text pairs.
- Image Editing: We use the dataset introduced in (Brooks et al., 2023), which provides input and target images along with corresponding editing instructions.
- Super Resolution: We use the Real-ESRGAN dataset (Wang et al., 2021), which consists of high-resolution images. We generate corresponding low-resolution images by applying degradations and downscaling them by half.
- Image Inpainting: We use the dataset from (Yildirim et al., 2023), which provides a multi-modal inpainting dataset designed for object removal based on text prompts.

More details can be found in Section A.1 in the Appendix.

**Training Details:** We freeze all other layers and train only the FFN experts, routers, and task-specific layers, as described in Section 5. This results in training 158M parameters for SDv1.5 and 1.5B parameters for SDXL. Both models are trained on 8× A100 GPUs for 100 epochs, with a batch size of 16 per GPU and image resolution of 512 × 512. SDXL is optimized using AdamW with a learning rate of 5e-5, while SDv1.5 is trained using Adam with a learning

rate of 1e-4. During sampling, we perform denoising for 20 and 50 iterations for multi-task SDv1.5 and SDXL models respectively. More details are presented in Section A.2 in the appendix.

**Metrics:** We used the following metrics that are commonly considered for evaluating models train for specific tasks.

- T2I: We report Frechet Inception Distance (FID) on the test set of the COCO captions dataset. Lower values indicated better images.
- SR: We measure Learned Perceptual Image Patch Similarity (LPIPS) (Zhang et al., 2018) between generated and ground truth images, where lower values indicate more similarity.
- Image Editing: We report Image-Text (I-T) Directional Similarity (Brooks et al., 2023), which quantifies how well the change in text captions aligns with corresponding edits. Let $I_{input}$ and $I_{edited}$ represent CLIP-extracted features of input and edited images respectively (Radford et al., 2021). Similarly, let $T_{edited}$ and $T_{input}$ denote the CLIP-extracted text features for the input and edited descriptions. I-T Directional Similarity is defined as $S(T_{edited} - T_{input}, I_{edited} - I_{input})$, where $S$ is the cosine similarity.
- Inpainting: We report Image-Image (I-I) Directional Similarity, which measures alignment with the ground truth. Given CLIP features corresponding to ground truth image $I_{gt}$, I-I Directional Similarity is defined as $S(I_{gt} - I_{input}, I_{edited} - I_{input})$. Higher values indicate better similarity to the ground truth.

## 7. Results

**Comparison with single-task and multi-task models**

We compare MTU against both single-task and multi-task baselines. For single-task baselines, we prioritize open-source models based on SDv1.5 or SDXL whenever available. If no open-source models are available, we fine-tune the pre-trained T2I model on the I2I task following the

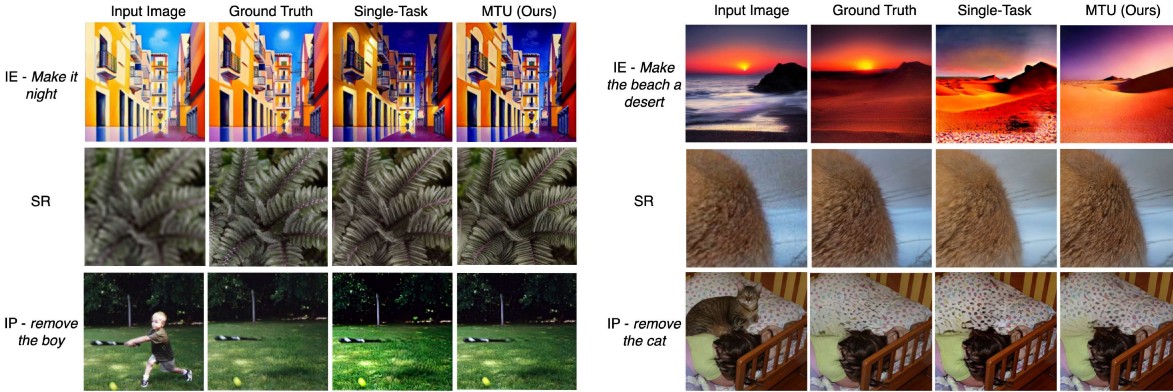

*Figure 4.* Qualitative comparison of MTU based on SDv1.5 (left) and SDXL (right) with corresponding single-task baselines for Image Editing (IE) (Brooks et al., 2023), Super Resolution (SR) (Rombach et al., 2022), and Inpainting (IP). (Yildirim et al., 2023)

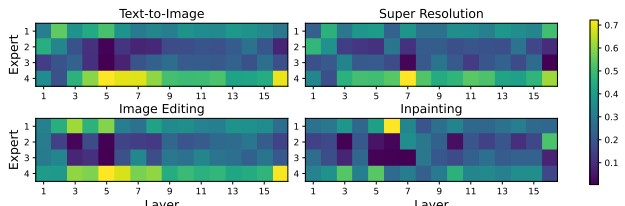

*Figure 5.* Analysis for expert selection by the router. We show the expert weight distribution assigned by the router for SDv1.5 with four experts (N = 4).

|  |  | IE | SR | IP |
|---|---|---|---|---|
|  |  | I-T Dir Sim ↑ | LPIPS ↓ | I-I Dir Sim ↑ |
| SDv1.5 | LORA | 8.3 | 43.3 | 31.4 |
|  | IA3 | 11.2 | 39.3 | 34.2 |
|  | Full - FT | 15.4 | 38.0 | 46.5 |
|  | MTU (N = 4) | **17.2** | **24.8** | **44.0** |
| SDXL | LORA | 13.1 | 31.8 | 28.9 |
|  | IA3 | 8.4 | 51.0 | 30.6 |
|  | Full - FT | 17.3 | 26.9 | 43.2 |
|  | MTU (N = 1) | **20.1** | **26.5** | **44.2** |

*Table 3.* Comparison with PEFT methods. MTU significantly outperforms PEFT methods applied on FFN layers.

methodology outlined by (Brooks et al., 2023) in Section 3. Specifically, we consider the image editing SDv1.5 and SDXL models fine-tuned by (Brooks et al., 2023). For Super-Resolution (SR) and Inpainting, we could only compare SDv1.5 based models from (Rombach et al., 2022) and (Yildirim et al., 2023), respectively. Since no equivalent open-source versions are available for SDXL, we fine-tuned a model locally for both SR and Inpainting.

Additionally, we use the Versatile Diffusion (VD) (Xu et al., 2023) and UniDiffuser (Bao et al., 2023) models as multi-task model baselines, as they are, to the best of our knowledge, the only models comparable to the MTU model. Note that these multi-task models were trained under entirely different settings and do not cover all the tasks considered in this paper. Therefore, for a fair comparison, we evaluate UniDiffuser solely on the T2I task and Versatile Diffusion on the T2I and Image Editing tasks.

Table 2 shows that our method consistently outperforms both single-task and multi-task baselines. Specifically, MTU based on SDv1.5 achieves scores of 17.2 in Image Editing and 24.8 in Super-Resolution (SR), while single-task baselines lag by 3 and 14 points, respectively. Additionally, MTU based on SDv1.5 outperforms existing multi-task baselines, further demonstrating its effectiveness. We also report the compute requirements for each model in GFLOPs, showing that our approach maintains computational efficiency

comparable to single-task models. Qualitative results are presented in Figure 4 and Section A.3 in the appendix.

**Comparison with PEFT methods**

Parameter-efficient fine-tuning (PEFT) methods are widely used to adapt pre-trained models to different tasks. Methods like LoRA (Hu et al., 2022) and IA3 (Liu et al., 2022) are lightweight, requiring only a small fraction of parameters to be fine-tuned within task-specific adapters. Since our analysis in Section 4 highlights that FFNs play a crucial role in enabling support for image-to-image tasks, we compare MTU with LoRA and IA3 for tuning the FFN layers. Note that applying IA3 only to FFN blocks did not yield good results, so we apply IA3 to the entire block including self-attention, cross-attention and FFN layers. By comparing our approach with these PEFT methods, we evaluate how effectively our method performs in comparison to efficient fine-tuning techniques that primarily target FFN layers. Table 3 compares MTU with PEFT methods such as LoRA and IA3. For SDXL, our method outperforms IA3 by approximately 6 absolute points in image editing and 15 absolute points in super-resolution (SR). Overall, MTU consistently outperforms LoRA and IA3 across all tasks, demonstrating better multi-task support compared to task-specific adapters.

**Analysing router assignment for tasks**

| | | # experts $N$ | top-$k$ | T2I | IE | SR | IP |
|---|---|---|---|---|---|---|---|
| | | | | FID ↓ | I-T Dir Sim ↑ | LPIPS ↓ | I-I Dir Sim ↑ |
| $(G = \frac{1}{N})$ | SDv1.5 | 1 | – | 7.3 | 17.6 | 25.3 | 44.0 |
| | | 2 | – | 8.4 | 9.0 | 30.9 | 43.2 |
| | | 4 | – | **7.2** | **17.2** | **24.8** | **44.0** |
| | | 8 | – | 8.0 | 10.2 | 27.8 | 38.7 |
| | | 8 | 4 | 7.9 | 8.9 | 25.8 | 39.3 |
| | | 16 | – | 8.5 | 9.6 | 27.2 | 36.5 |
| | | 16 | 4 | 8.2 | 8.6 | 26.5 | 38.6 |
| | SDXL | 1 | – | **3.9** | **20.1** | **26.5** | **44.2** |
| | | 2 | – | 10.5 | 10.4 | 30.4 | 39.9 |
| | | 4 | – | 12.3 | 11.8 | 30.5 | 38.6 |
| | | 8 | – | 12.8 | 12.4 | 31.6 | 34.5 |
| | | 8 | 4 | 13.6 | 12.8 | 31.5 | 32.7 |
| | | 16 | – | 17.1 | 11.8 | 32.8 | 31.7 |
| | | 16 | 4 | 18.2 | 10.9 | 33.2 | 31.9 |

| | | # experts $N$ | $G$ | # Params | T2I | IE | SR | IP |
|---|---|---|---|---|---|---|---|---|
| | | | | | FID ↓ | I-T Dir Sim ↑ | LPIPS ↓ | I-I Dir Sim ↑ |
| SDXL (2.6B) | $G = 1$ | 1 | 1 | 2.6B | 3.9 | **20.1** | 26.5 | 44.2 |
| | | 2 | 1 | 3.5B | **3.8** | **20.0** | 26.3 | 46.9 |
| | | 4 | 1 | 5.2B | **3.8** | 19.1 | **25.8** | **49.3** |
| | $G < 1$ | 4 | 0.25 | 2.6B | 12.3 | 11.8 | 30.5 | 38.6 |
| | | 2 | 0.5 | 2.6B | **10.5** | 10.4 | 30.4 | 39.9 |
| | | 4 | 0.5 | 3.5B | 11.3 | **12.3** | **28.6** | **40.9** |
| | $G > 1$ | 1 | 2 | 3.5B | 6.3 | 19.8 | 26.6 | 44.5 |
| | | 2 | 2 | 5.2B | **5.8** | **20.4** | **24.8** | **44.7** |
| | | 1 | 4 | 5.2B | 13.7 | 18.9 | 30.2 | 36.7 |

*Table 4.* **Top:** Performance of MTU on SDv1.5 and SDXL with varying number of experts $N$ and parameter constraint by setting $G = \frac{1}{N}$. **Bottom:** Performance of MTU models (SDXL) without parameter constraint. We allow model size to scale with the number of experts to show that multi-task performance is linked to both number of experts $N$ and the capacity of each expert $G$.

We analyze how the router distributes weights across the set of experts for each task to determine which experts specialize in specific tasks. Figure 5 illustrates the expert weight distribution assigned by the router for SDv1.5 with $N = 4$. Our findings show that Text-to-Image (T2I) generation and Image Editing share three experts: $E_4^5$, $E_4^6$, and $E_4^{16}$. Meanwhile, $E_4^7$ specializes in Super-Resolution (SR), and $E_1^6$ is assigned greater importance for Image Inpainting. The weight distribution across experts is similar for T2I (trained on COCO) and Image Editing, as both tasks require strong prompt-following capabilities. In contrast, Inpainting primarily relies on object removal prompts, while SR operates without any textual conditioning, necessitating the use of different experts for these tasks.

**MTU with varying the number of experts and sizes**

We perform two ablation studies (i) Determine the optimal number of experts $N$ while constraining the total parameter count of the MTU model as described in Section 5. (ii) Remove this constraint by varying $N$ and expert size $d_{\text{expert}}$ to evaluate how the number of experts and their capacity affect multi-task performance. We denote the ratio of an expert's dimension in MTU to the FFN dimension of the original model as $G = d_{\text{expert}}/d_{\text{ffn}}$.

For (i), we set $G = \frac{1}{N}$ such that the hidden dimension of each expert decreases as we increase the number of experts. Table 4 (Top) presents the performance of our method with varying number of experts under this constraint. Our findings show that for both SDv1.5 and SDXL, increasing the experts to a higher number (smaller experts) significantly degrades performance. For SDv1.5, the optimal number of experts is N = 4, while for SDXL, dividing the model into experts is suboptimal, as tuning the task-specific layer norms and FFN blocks alone provide sufficient multi-task support. Given that MTU for SDXL works best with a single expert, we conducted an additional experiment where FFNs were frozen, and only the layer norms preceding them were fine-tuned. However, this model failed to converge, implying that FFNs play a key role in learning I2I tasks. An alternative is to increase experts to 8 or 16, but select only the top 4 per task, keeping active parameters equal to the original FFN. Table 4 (Top) shows that this strategy underperforms compared to using all experts directly.

For (ii), we lift the parameter constraint from (i) and vary $G$ from $\frac{1}{N}$ to 1 in Table 4 (Bottom). *Increasing the number of experts enhances performance across most tasks*, particularly super-resolution (SR) and image inpainting (IP). For instance, in the case of $G = 1$ (experts matching the pre-trained FFN size), the MTU model with $N = 2$ or $N = 4$ outperforms the $N = 1$ baseline (Table 2). When $G < 1$, the configuration with $N = 4$ consistently outperforms $N = 2$ for the same value of $G$.

For a fixed number of experts, the configuration with $G = 1$ (expert size equal to the pre-trained FFN) consistently outperforms both $G < 1$ (smaller experts) and $G > 1$ (larger experts), with $G < 1$ yielding the weakest results. Interestingly, Table 4 (top) shows the opposite trend for SDv1.5, where smaller experts lead to better performance. *This suggests that multi-task performance is influenced by both expert capacity (width) and model depth.* In shallower models like SDv1.5, reducing expert width is advantageous, while in deeper architectures such as SDXL, maintaining the original expert dimensions yields better outcomes. This is likely because modifying expert size in a deep network, without proportionally adjusting the rest of the architecture, disrupts training stability and degrades overall performance.

In summary, multi-task performance depends on three interrelated factors: expert capacity, expert count, and model depth. With a fixed expert size, increasing the number of experts generally boosts performance. Shallow architectures like SDv1.5 benefit from smaller experts, while deeper models such as SDXL perform best when experts match the pre-trained FFN size. This occurs because, in very deep networks, changing expert dimensions without adjusting the surrounding layers can destabilize training and degrade quality. In Section 8, we provide a practical guideline for selecting the optimal expert size and count based on available compute resources.

**Exploring task interference within MTU framework**

| | Tasks | T2I | IE | SR | IP |
|---|---|---|---|---|---|
| | | FID ↓ | I-T Dir Sim ↑ | LPIPS ↓ | I-I Dir Sim ↑ |
| $T = 1$ | T2I | 12.9 | – | – | – |
| | IE | – | 15.4 | – | – |
| | SR | – | – | 29.3 | – |
| | IP | – | – | – | 46.5 |
| $T = 2$ | T2I, IP | 17.9 | – | – | 30.0 |
| | SR, IP | – | – | 36.2 | 30.2 |
| | IE, IP | – | 20.6 | – | 52.6 |
| | T2I, SR | 13.4 | – | 22.3 | – |
| | IE, SR | – | 17.1 | 24.6 | – |
| | T2I, IE | 6.9 | 18.5 | – | – |
| $T = 3$ | T2I, IE, SR | 7.0 | 18.0 | 22.7 | – |
| | IE, SR, IP | – | 16.8 | 25.1 | 35.8 |
| | T2I, IE, IP | 13.5 | 17.1 | – | 45.2 |
| $T = 4$ | T2I, IE, SR, IP | 7.2 | 17.2 | 24.8 | 44.0 |

*Table 5.* Exploring Task interference by training the MTU model across all combinations of the four tasks.

Table 5 illustrates how different tasks interact and interfere with each other within the MTU framework. We train the MTU model across all combinations of the four tasks to analyze how the inclusion of one task impacts the performance of another. We denote by $T$ the number of tasks included in each experiment (with $T = 1$ for single-task training and $T = 4$ when all tasks are included). For intermediate values of $T$, we explicitly specify which tasks are involved.

From $T = 2$ experiments, we first observe that image editing (IE) and image inpainting (IP) are highly compatible tasks, with performance increasing from $44.0$ in our MTU model to $52.6$ when combined. In contrast, super-resolution (SR) and text-to-image (T2I) appear to be less compatible with IP, likely because T2I and SR require generating new objects, whereas IP focuses on removing objects. Notably, the compatibility between IP and IE may stem from the fact that the IP dataset (InstructPix2pix) includes editing instructions that also involve object removal. We observe that T2I and IE are highly compatible, as training them together improves IE performance—an effect also highlighted in Figure 5, where these tasks select the same experts. For $T = 3$, adding IP to SR and IE hurts SR performance, and training T2I, IE, and IP together also reduces T2I despite strong IP–IE compatibility. In the full $T = 4$ setting, T2I, IE, and SR improve over their single-task baselines, while IP drops by 2 points.

Our results highlight notable task interference, and we recommend that future research adapt proven multi-task learning techniques, such as gradient conflict mitigation (Zhang et al., 2024c), or dynamic loss balancing (Navon et al., 2022) to alleviate interference in diffusion models.

## 8. Discussions

In this section, we provide recommendations for upcycling a pre-trained text-to-image model into a multi-task image generation model. Until now, PEFT methods are often the first choice for practitioners seeking to enable multi-task support due to their computational efficiency. However, our experiments with task-specific PEFT methods show that while they are lightweight, our MTU approach consistently outperforms them across all tasks while maintaining the same computational budget. We also demonstrate that MTU models can be tailored to any compute budget by adjusting expert sizes, making MTU an ideal solution for resource-constrained multi-task applications. Based on our findings, we propose an improved recipe for enabling multi-task capabilities in pre-trained models based on a fixed compute budget. We recommend determining your total parameter budget first.

- Start by adding task-specific input convolution layers to process additional image conditioning. Introduce task-specific layer norms before FFNs, and fine-tune the model without splitting any FFN into smaller experts (i.e., keeping a single expert). As shown in Table 4, this simple approach performs well for SDXL but is less effective for SDv1.5.

- If a single FFN underperforms, conduct an ablation by varying expert count until the total parameters fit your budget. Under tighter limits, increase expert count and proportionally shrink each expert's hidden size. If resources allow a larger model, keep each expert at the pre-trained FFN size and add more experts for optimal multi-task performance.

## 9. Conclusions

In this work we introduced Multi-task Upcycling, a simple yet effective approach to enhance pre-trained text-to-image models, such as SDv1.5 and SDXL, to support multiple image editing tasks. Unlike previous approaches, MTU is the first multi-task diffusion modeling framework that seamlessly integrates multi-task learning with on-device compatibility, ensuring efficiency without compromising performance. Our idea is based on an empirical observation that parameters in FFN layers in diffusion models deviate the most during task-specific fine-tuning. We use this observation to propose splitting of existing FFN layer into smaller FFN experts, which are then combined with a router network. We conduct an extensive evaluation across Text-to-Image, Image Editing, Super-Resolution, and Inpainting tasks, demonstrating superior performance compared to both single-task and multi-task baselines. We believe that our approach will open up new avenues of research in the rapidly evolving area of image synthesis and will continue helping the efforts in making multi-task vision models efficient for on-device deployment.

## Impact Statement

This paper presents work in the field of Multi-Task image generation models. There are many potential societal consequences of our work, none which we feel must be specifically highlighted here.

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

# A. Appendix

## A.1. Dataset details

In this section, we provide details on the tasks and datasets used for training MTU models. Our study incorporates the following datasets, with the exact data splits outlined in Table 6.

- Text-to-Image Generation (T2I): We utilize the COCO Captions dataset (Lin et al., 2015), a large-scale collection of image-text pairs. Each image in this dataset is accompanied by five captions, with one randomly selected during training.
- Image Editing: We use the dataset introduced in (Brooks et al., 2023), which includes input-target image pairs along with corresponding editing instructions. Each input image has 4-5 target variations for a given edit instruction, with one randomly selected during training.
- Super Resolution: We use the Real-ESRGAN dataset (Wang et al., 2021), which consists of high-resolution images. We generate corresponding low-resolution images by applying degradations like Poisson and Gaussian blur and downscaling them by half. For SR, we input an empty string to the model.
- Image Inpainting: We utilize the dataset from (Yildirim et al., 2023), a multi-modal inpainting dataset designed for object removal based on text prompts. Built on the GQA dataset (Hudson & Manning, 2019), it leverages scene graphs to generate paired training data using state-of-the-art instance segmentation and inpainting techniques.

## A.2. Training and Inference Details

In this section, we present training and inference details for both single-task and MTU models.

**Training details:** Both single-task and MTU models trained on 8× A100 GPUs for 100 epochs, with a batch size of 16 per GPU and image resolution of $512 \times 512$. SDXL is optimized using AdamW with a learning rate of 5e-5, while SDv1.5 is trained using Adam with a learning rate of 1e-4. For SDXL, we find that using a weight decay of 0.01 helps stabilize training.

**Inference details:** During sampling, we perform denoising for 20 iterations in multi-task SDv1.5 and 50 iterations in SDXL. For Text-to-Image (T2I) generation, Image Editing, and Inpainting, we apply Classifier-Free Guidance (CFG) (Ho & Salimans, 2022). However, for Super-Resolution (SR), no CFG is used, as it only processes an empty string as input. For T2I generation, we use a guidance scale of 7.5 for SDv1.5 and 5.0 for SDXL. For Image Editing and Inpainting, we follow the CFG strategy from (Brooks et al., 2023), which employs dual guidance scales—one for image and another for text. For SDv1.5, we use an image guidance scale of 1.6 and a text guidance scale of 7.5 for Image Editing, and 1.5 and 4.0 for Inpainting, respectively. For SDXL, we set the image guidance scale to 1.5 and the text guidance scale to 10.0 for Image Editing, while for Inpainting, we use 1.5 for image and 4.0 for text. For more details on the formulation of CFG, we direct the readers to (Brooks et al., 2023).

| Task | Dataset | Train | Val | Test |
|---|---|---|---|---|
| T2I | COCO Captions (Li et al., 2017) | 118287 | 5000 | 5000 |
| Image Editing | InstructPix2pix (Brooks et al., 2023) | 281709 | 31301 | 2000 |
| Super Resolution | Real- ESRGAN(Wang et al., 2021) | 23744 | 100 | 100 |
| Inpainting | GQA-Inpaint (Yildirim et al., 2023) | 90089 | 10009 | 5553 |

*Table 6.* Training, validation and test splits of the datasets used in training MTU

## A.3. Qualitative Results

We provide more qualitative results for each of the tasks considered in the paper from Figures 7, 8, 9, and 10.

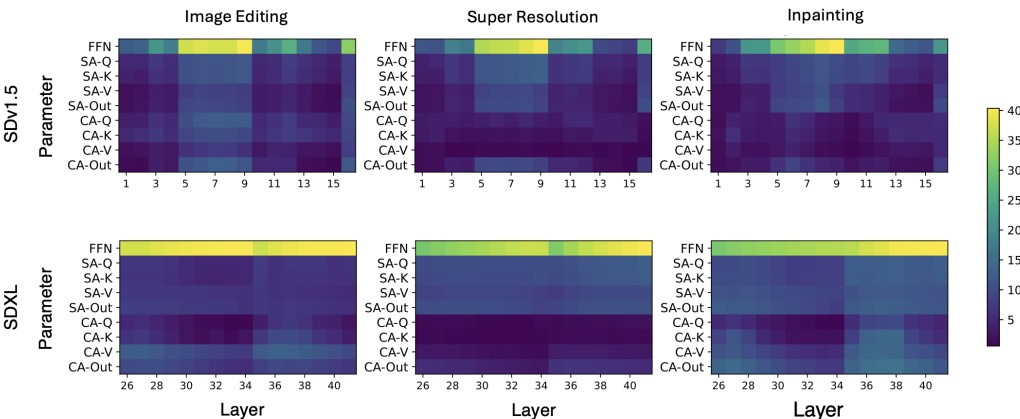

*Figure 6.* Distances between fine-tuned weights $\theta_f^\tau$ and pre-trained initialization $\theta_p$ for SDv1.5 (top) and SDXL (bottom) across Image Editing, Super-Resolution, and Inpainting. Here x-axis corresponds to the Layer index and the y-axis corresponds to the distance between the parameters. For all tasks, FFNs exhibit the highest deviation from initialization, highlighting their crucial role in adapting to downstream tasks. In SDXL, we focus only on the middle layers, where this deviation is most pronounced.

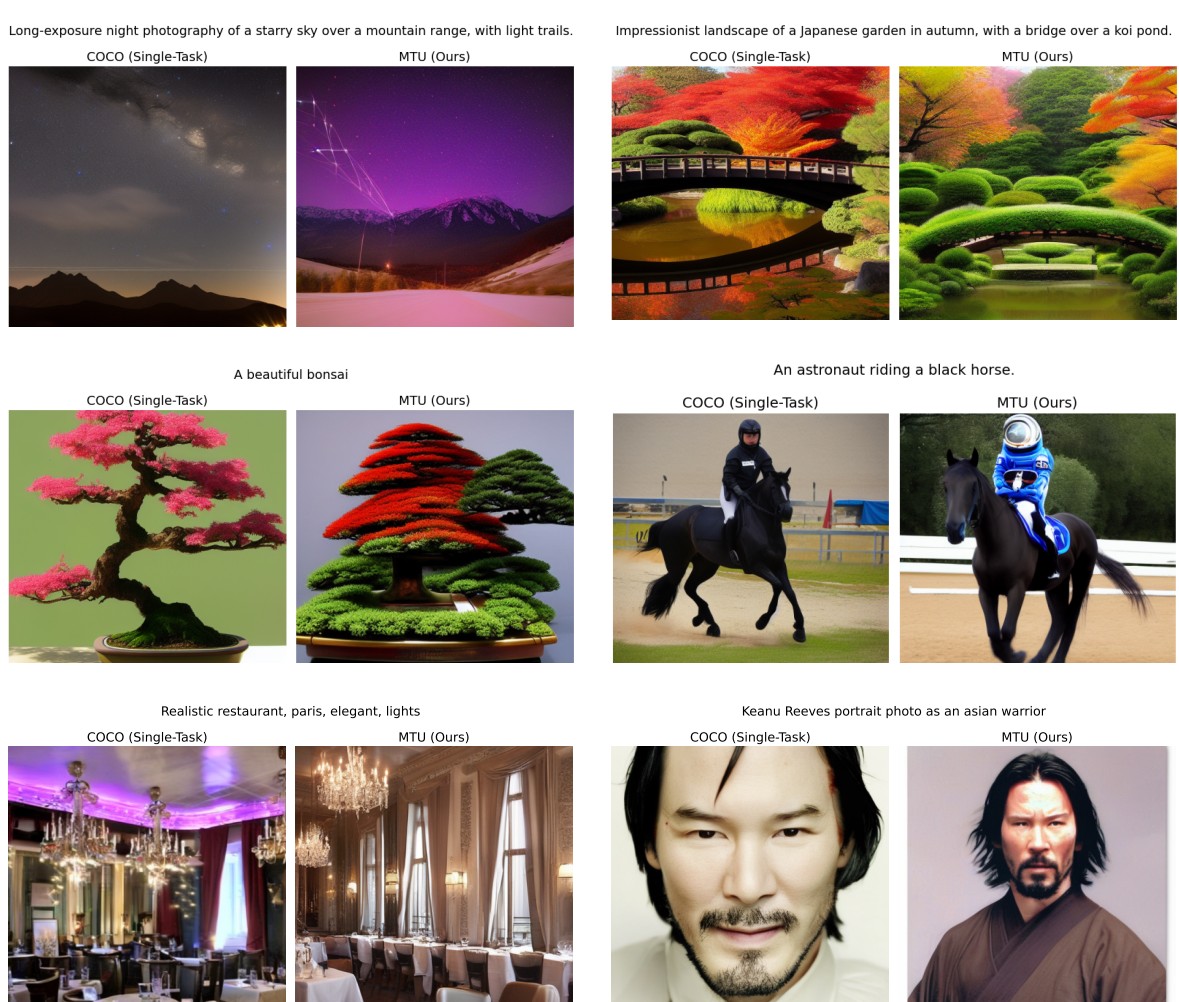

*Figure 7.* Qualitative results of MTU based on SDv1.5 (first two rows) and SDXL (bottom row) for Text-to-Image Generation.

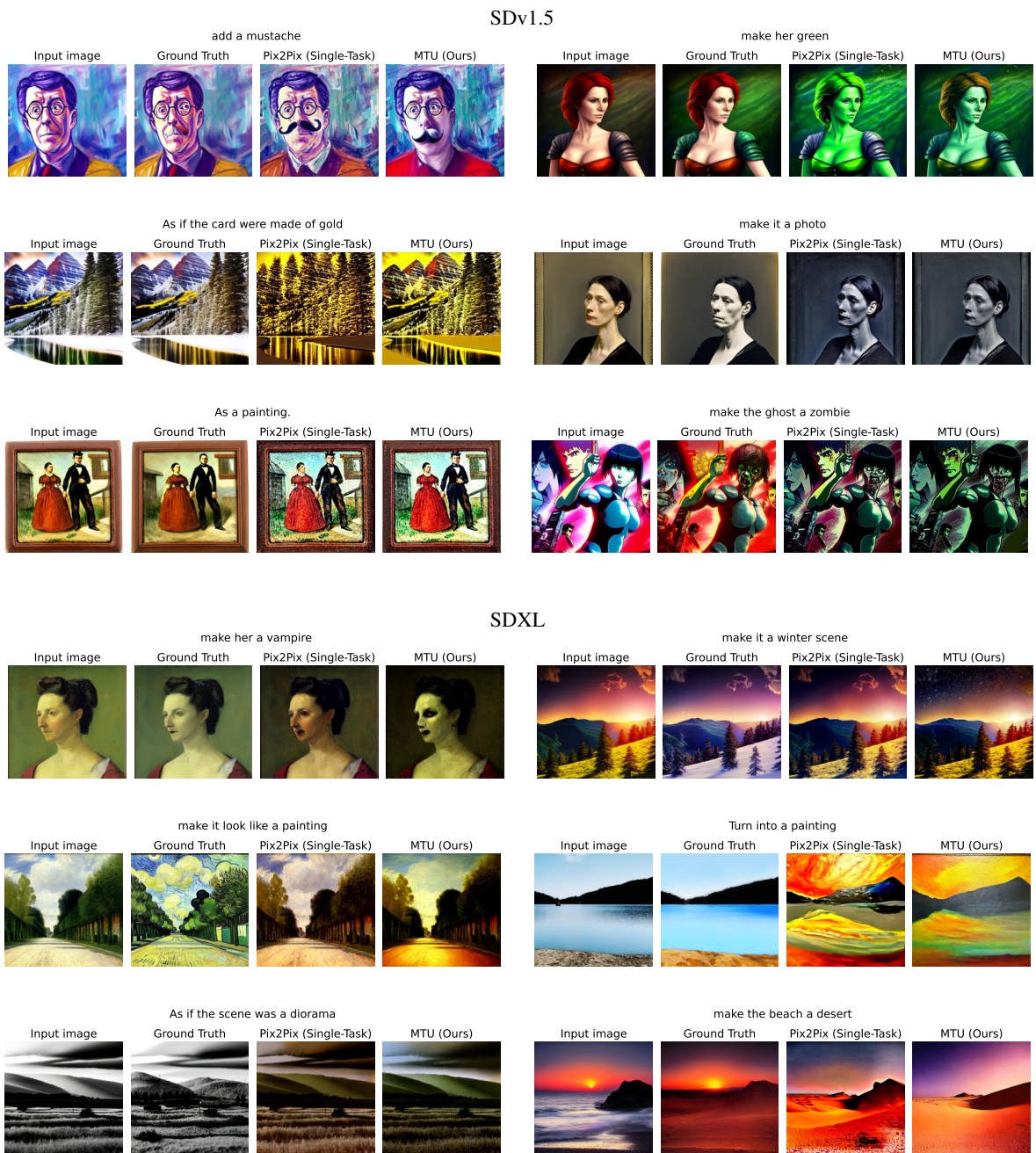

*Figure 8.* Qualitative results of MTU based on SDv1.5 and SDXL for Image Editing. In both models, our approach produces high-quality images with superior prompt adherence and faithful edits.

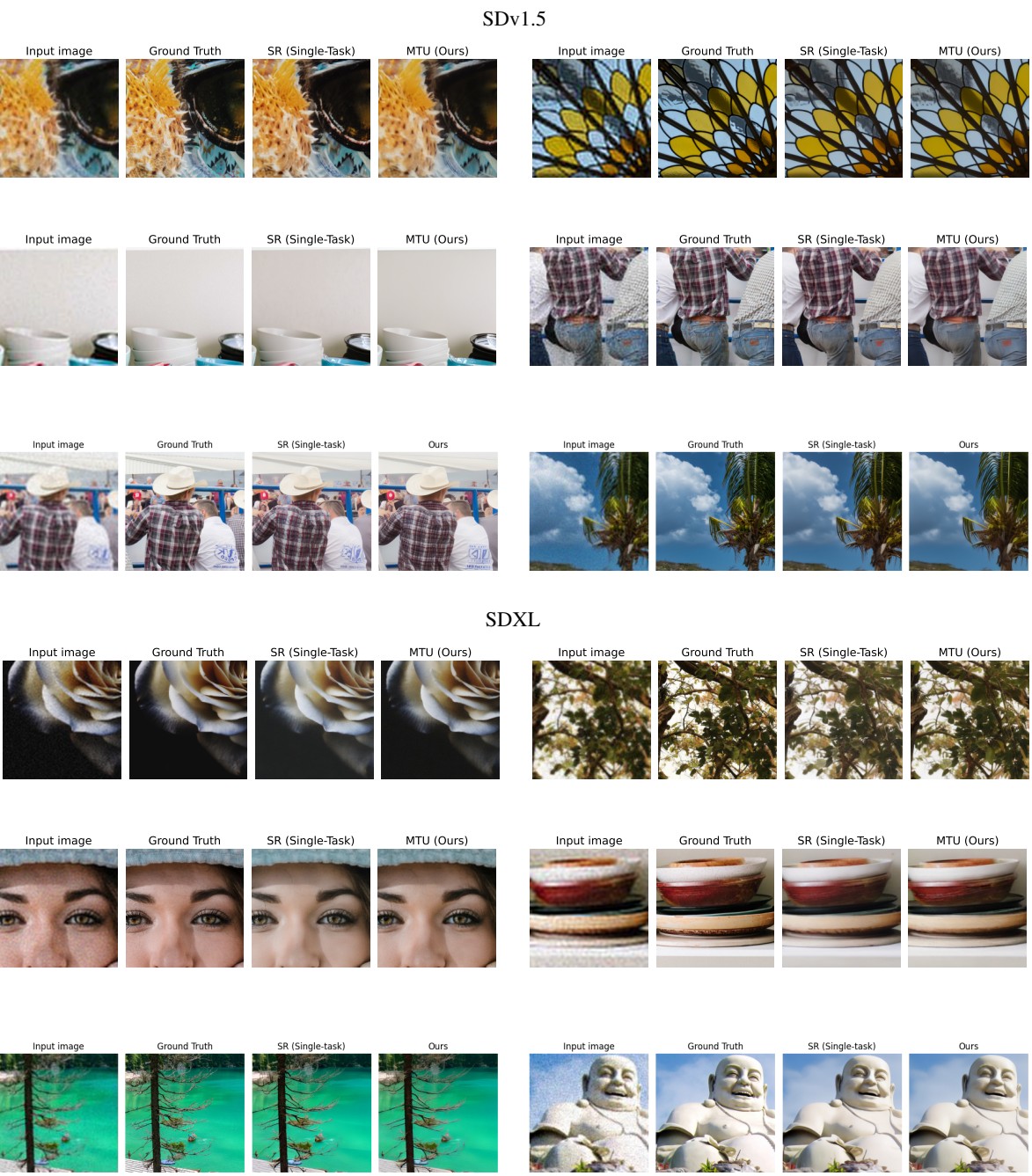

*Figure 9.* Qualitative results of MTU based on SDv1.5 and SDXL for Super Resolution. Our method effectively restores high-resolution images from low-resolution inputs that have been corrupted by image degradations, producing clearer and more detailed outputs.

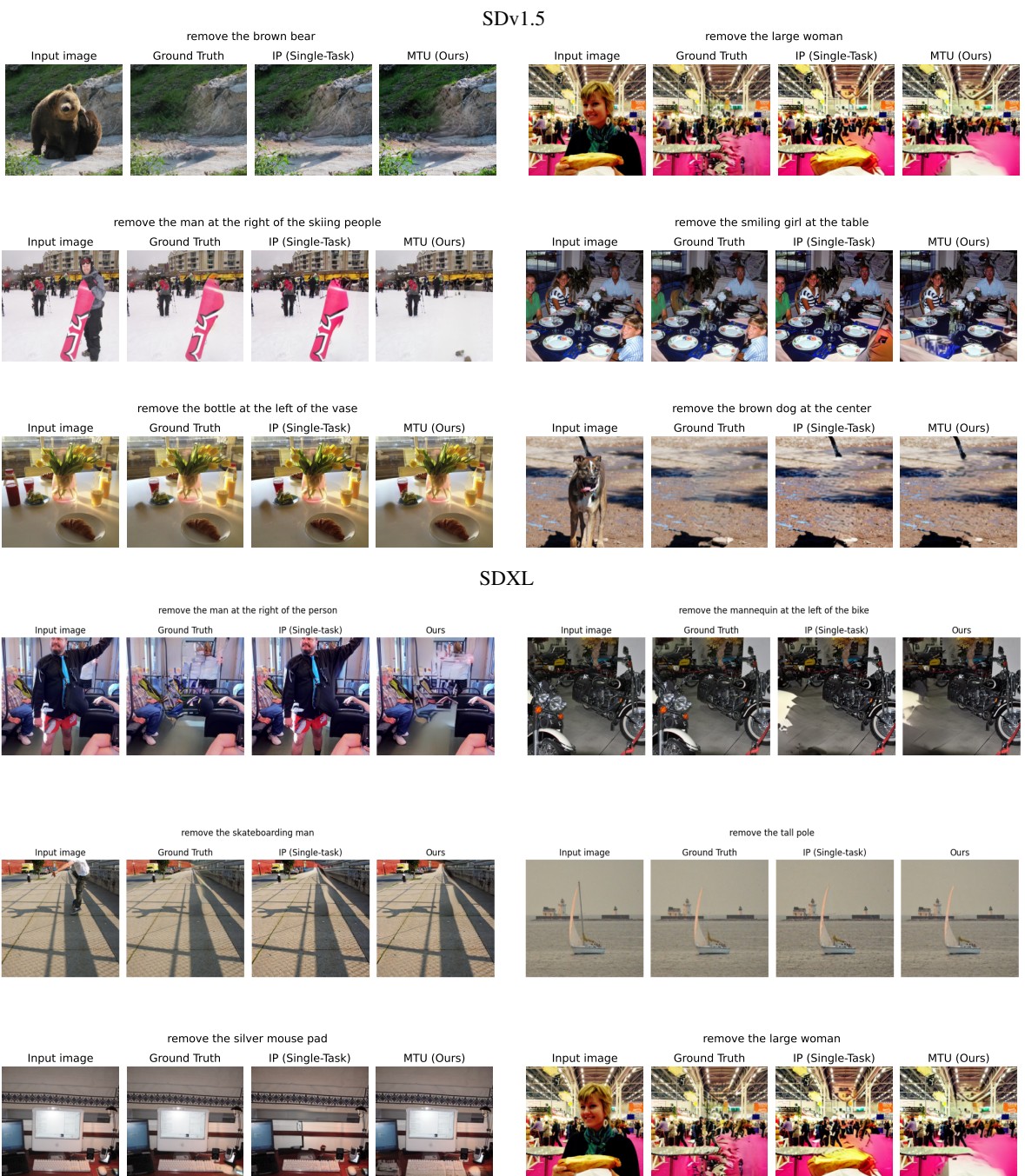

*Figure 10.* Qualitative results of MTU based on SDv1.5 and SDXL for Image Inpainting. Our method demonstrates better object removal, generating clean inpainted images.

