# OpenReview forum: "Upcycling Text-to-Image Diffusion Models for Multi-Task Capabilities"
_ICML.cc/2025/Conference — ICML 2025 poster_

### Official Review · Reviewer_FqBN · 2025-03-02

**Overall Recommendation:** 3

**Summary:**

This paper proposes Multi-Task Upcycling (MTU), a method to extend pre-trained text-to-image diffusion models for multiple image-to-image generation tasks without significantly increasing computational complexity or model parameters. The key idea is to replace the Feed-Forward Network (FFN) layers with smaller FFN experts combined through a dynamic routing mechanism guided by task-specific embeddings. This approach enables the model to support tasks like image editing, super-resolution, and inpainting, while maintaining comparable performance to single-task fine-tuned models with similar computational costs. The MTU method is inspired by Mixture-of-Experts (MoE) models commonly used in language models and adapts them for diffusion models.

**Claims And Evidence:**

The claims made in the submission are supported by clear and convincing evidence.

**Essential References Not Discussed:**

This paper includes the work related to the key contributions.

**Experimental Designs Or Analyses:**

The experimental design is comprehensive, with ablation studies on the number of experts, routing mechanisms, and comparisons to PEFT methods. However, the paper could benefit from analyzing the optimal number of experts (N=4 for SDv1.5 and N=1 for SDXL).

**Methods And Evaluation Criteria:**

The method is well-motivated and evaluation criteria make sense for the image-to-image generation tasks. However, some qualitative examples could better highlight failure cases, especially in challenging scenarios.

**Other Comments Or Suggestions:**

No other comments or suggestions.

**Other Strengths And Weaknesses:**

Strengths:
1. The MTU framework presents a simple yet effective design that minimizes parameter inflation while preserving the model’s multi-task capabilities.
2. The method is well-suited for on-device deployment, addressing an important challenge of diffusion models in resource-constrained environments.
3. The paper includes comprehensive ablation studies.

Weaknesses:
1. The paper provides limited analysis of failure cases, which would help better understand the limitations of the proposed method across diverse tasks.
2. The optimal number of experts is not rigorously analyzed. For example, the observation that MTU with 4 experts outperforms MTU with 1 expert on SDv1.5 lacks a clear explanation.
3. The significant performance degradation of MTU on SDXL with 4 experts compared to 1 expert raises concerns about scalability. Further analysis is needed to clarify why increasing the number of experts negatively impacts performance on larger models.
4. The paper does not explore task interference effects, which could potentially affect the model’s performance when handling multiple tasks simultaneously. Including an analysis of how the presence of certain tasks impacts others would strengthen the evaluation.

**Questions For Authors:**

See Other Strengths And Weaknesses.

**Relation To Broader Scientific Literature:**

The paper is well-positioned in the multi-task diffusion models and Mixture-of-Experts (MoE) literature, and promotes the research on multi-task diffusion modeling methods.

**Theoretical Claims:**

No significant novel theoretical claims are made. The routing mechanism and expert combination follow the concept of fine-grained MoE models in the LLM literature and adapt it for multi-tasking in diffusion models.

---

> ### Author Rebuttal · Authors · 2025-03-31
>
> We thank you for your insightful feedback. We answer your concerns below.
>
> **Concerns over scalability of experts in SDXL:**  We acknowledge your concern regarding the performance drop observed when increasing the number of experts in SDXL —a point also raised by Reviewer ptLD. In our rebuttal for Reviewer ptLD, we demonstrate that multi-task performance is linked to the width (i.e., expert capacity) and depth of the model. Specifically, reducing the width of the experts in shallow models like SDv1.5 works well, however in the case of deeper models like SDXL, larger experts perform well.
>
> We point out that in Tables 1 and 4 of the main paper, we maintained the overall model size by reducing the hidden dimensions of the experts as their number increased. However, when this constraint is removed (as shown in Table A of ptLD's rebuttal), increasing the number of experts in our MoE setup significantly enhances multi-task performance while remaining more efficient than deploying separate models for each task. These results address your concerns regarding scalability and align with our goal of avoiding parameter inflation as the number of tasks grows.
>
> **Exploring task interference:** We thank the reviewer for raising this important point. In the table below, we present an analysis of how different tasks interact by training various combinations of the four tasks considered in our paper. For this analysis, we trained SDv1.5 with $N=4, G=0.25$ (where $G$ is the ratio of an expert's dimension in MTU to the FFN dimension of the original model) across all task combinations, and we also report the performance of single-task models trained without interference. (Note: In the main paper, the performance of the super-resolution (SR) model is reported for an open-source version that uses image captions to generate high-resolution images. For a fair comparison and to analyze task interference, we trained an SR model with an empty string as the text condition and report its performance accordingly. Please look at Section A.2 in the paper.)
>
> | Tasks       | T2I       | IE          | SR       | IP          |
> |-------------|-----------|-------------|----------|-------------|
> |             | FID ↓     | I-T Dir Sim ↑ | LPIPS ↓ | I-I Dir Sim ↑ |
> | Single-task | 12.9      | 15.4        | 29.3     | 46.5        |
> | T2I-IP      | 17.9      | --          | --       | 30.0        |
> | T2I-IE      | 6.9       | 18.5        | --       | --          |
> | T2I-SR      | 13.4      | --          | 22.3     | --          |
> | SR-IP       | --        | --          | 36.2     | 30.2        |
> | SR-IE       | --        | 17.1        | 24.6     | --          |
> | IE-IP       | --        | 20.6        | --       | 52.6        |
> | T2I-IE-SR   | 7.0       | 18.0        | 22.7     | --          |
> | T2I-IE-IP   | 13.5      | 17.1        | --       | 45.2        |
> | IE-SR-IP    | --        | 16.8        | 25.1     | 35.8        |
> | All      | 7.2       | 17.2        | 24.8     | 44.0        |
>
> Our summary of task interference from the above table -
>
> - First, we observe that image editing (IE) and image inpainting (IP) are highly compatible tasks, with performance increasing from 44.0 in our MTU model to 52.6 when combined. In contrast, super-resolution (SR) and text-to-image (T2I) appear to be less compatible with IP, likely because T2I and SR require generating new objects, whereas IP focuses on removing objects. Notably, the compatibility between IP and IE may stem from the fact that the IE dataset (InstructPix2pix) includes editing instructions that also involve object removal.
> - We observe that T2I and IE are highly compatible, as training them together improves IE performance—an effect also highlighted in Figure 5 in the main paper, where these tasks select the same experts. While SR and IE may be compatible when trained together, adding IP into the mix leads to a significant drop in SR performance. Moreover, when T2I, IE, and IP are trained concurrently, T2I performance declines; however, the strong compatibility between IP and IE ensures that their performance remains high.
> - In our multi-task model, where all tasks are trained jointly, we observe that the performance of the T2I, IE, and SR tasks improves compared to their single-task baselines, while performance decreases for IP by 2 points. This indicates that some degree of task interference may be prevalent.
>
> We appreciate the reviewer for highlighting this issue. However, we view reducing task interference as an important direction for future work and will address this as a limitation in the revised version of the paper.

---

### Official Review · Reviewer_ptLD · 2025-03-12

**Overall Recommendation:** 3

**Summary:**

This paper suggests that upcycling the text-to-image (T2I) diffusion model when adapting it on the multi-task learning. In detail, when fine-tuning T2I diffusion mode with multiple tasks (e.g. SR, Inpainting, T2I generation, Image editing), they argue to utilize Mixture-of-Experts (MoE). For the motivation, they observe that the fine-tuning diffusion model on a specific task leads to significant changes in FFN parameters compared to the other modules such as attention layers. Thus, they divide FFN into multiple experts with a task-specific layer normalization. Experimental results show that the proposed Multi-Task Upcycling (MTU) achieves a better performance compared to other PEFT methods. Notably, they also show that the model w/ MTU surpasses a single expert model fine-tuned on a single task.

**Claims And Evidence:**

- They claim that when applying multi-task learning to the diffusion model (i.e. training diffusion model on several image generation tasks), we need to use multi-task upcycling.

- As evidence, they show the experiment that the parameters of FFN change mostly when fine-tuning the diffusion model to a specific task, and achieving the highest performance compared to the models only training with SA or CA.

- Experimental results also support their claim, as diffusion model trained on multiple tasks with/ MTU surpasses a single expert model on a specific task.

- One major concern is the experiment in Table. 4. It shows a single expert is enough in most cases, thus it degrades the significance of the author's claim, the importance of MoE architecture (instead, a just task-specific normalization is sufficient for multi-task learning in DM). Also, 2 experts in SDv1.5 reports much worse results compared to (1, 4) expert models, and I wonder why 2 experts have such worse performance.

**Essential References Not Discussed:**

N/A

**Experimental Designs Or Analyses:**

- The main table (Table. 2) shows the superiority of the proposed method by suggesting the performance enhancement of the diffusion model trained with multi-tasks surpasses the performance of a task-specific model.

- Figure 5. shows that the different experts are activated when the different types of tasks are given.

- Table 3. shows the significance of the proposed method over the typical PEFT methods such as LoRA.

**Methods And Evaluation Criteria:**

- The proposed methods seem to make sense. They introduce a task-specific normalization and task-specific MoE and task-specific input layers. Input layer should be adaptive as the input image would differ as they are trained on different tasks. Also, as they show the FFN mostly changes during fine-tuning the diffusion model, introducing the task-specific normalization and MoE also seems to work well.

- Evaluation metrics are common metrics for each task. For example, they use FID for text-to-image image generation and LPIPS for SR task. These are the most common metrics to measure the generative and perceptual SR capability.

**Other Comments Or Suggestions:**

N/A

**Other Strengths And Weaknesses:**

N/A

**Questions For Authors:**

As I mentioned, the major issue for me is that the significance of MoE.
In most cases, a single expert is sufficient to achieve the best performance.
Then, why do we need to use MoE instead of just fine-tuning the FFN layer with a task-specific normalization?

**Relation To Broader Scientific Literature:**

The proposed method suggests the technique to enhance the fine-tuning of the diffusion model on multiple tasks. It would be extended to the other tasks like ControlNet, which has multiple types of inputs as conditions.

**Theoretical Claims:**

There are no theoretical claims in this paper.

---

> ### Author Rebuttal · Authors · 2025-03-31
>
> Thank you for your insightful review. We acknowledge that your concerns mainly pertain to Table 4, where we present an ablation study on the number of experts for both SDv1.5 and SDXL. We understand the reviewer is concerned regarding the significance of our method, given that in many cases a single expert ($N=1$) appears to perform adequately. We address your concerns below.
>
> **Experimental setup in Table 4:**  Our paper aims to maintain a parameter count similar to that of the original pre-trained model. To achieve this, we reduce the hidden dimension of each expert as the number of experts increases, ensuring that the ratio of an expert's dimension in MTU to the FFN dimension of the original model, remains $G = \frac{1}{N}$.
>
>  In Table 4 of the paper, we conducted an ablation study over $N$ (and consequently $G$) to determine the optimal configuration for both SDv1.5 and SDXL. Our results indicate that for SDv1.5, the best configuration is $N=4, G=0.25$. In contrast, for SDXL, the optimal performance is achieved with $N=1, G=1$, suggesting that retaining the original FFN size, with the addition of task-specific layer norms, is sufficient for handling multiple tasks in SDXL. Following this, we conduct an ablation over the size of the experts by removing the parameter constraint in Table 4.
>
> **Ablation over size of the experts:** We hypothesize that for a given model, multi-task performance depends more on the size of the FFNs than on the number of experts. To test this, we removed the parameter constraint and performed an ablation study by varying $G$ from $\frac{1}{N}$ to 1. Table A below presents these results for SDXL, revealing that performance on tasks, particularly IE, SR, and IP, improves as the expert dimension increases. For instance, when $N=2$ with $G=1$ and $N=4$ with $G=1$, the performance in SR and IP improves significantly compared to the model reported in the paper $N=1, G=1$. These findings support our hypothesis. We also see that for a fixed value of $G$, increasing the number of experts improves performance for image-image tasks. Therefore, the MoE architecture becomes especially significant when scaling the model size for SDXL.
>
> Table A: Ablation over number of experts for SDXL
>
> |          | Single-task |   | $N=1$   |   | $N=2$   | $N=2$   |   | $N=4$    | $N=4$    | $N=4$   |
> |----------|-------------|---|---------|---|---------|---------|---|----------|----------|---------|
> |          |             |   | $G=1$   |   | $G=0.5$ | $G=1$   |   | $G=0.25$ | $G=0.5$  | $G=1$   |
> | #Params  | 2.6B        |   | 2.6B    |   | 2.6B    | 3.5B    |   | 2.6B     | 3.5B     | 5.2B    |
> | T2I   (FID $\downarrow$)   | 4.1         |   | 3.9     |   | 10.5    | 3.8     |   | 12.3     | 11.3     | 3.8     |
> | IE    (I-T Dir Sim $\uparrow$)   | 17.3        |   | 20.1    |   | 10.4    | 20.0    |   | 11.8     | 12.3     | 19.1    |
> | SR  (LPIPS $\downarrow$ )   | 26.9        |   | 26.5    |   | 30.4    | 26.3    |   | 30.5     | 28.6     | 25.8    |
> | IP   (I-I Dir Sim $\uparrow$)    | 43.2        |   | 44.2    |   | 39.9    | 46.9    |   | 38.6     | 40.9     | 49.3    |
>
> **Analysing the ablation over number of experts for SDv1.5 (Table 4):** For SDv1.5, we determined that the optimal configuration is $N=4$ and $G=0.25$. Models using $N=1, G=1$ occasionally produced unusual artifacts in SR, and while we do not fully understand the cause, the $N=4, G=0.25$ setup consistently avoided these issues. Additionally, training became unstable with $N=8$ or $N=16$ due to the excessive reduction in expert size. The $N=2, G=0.5$ configuration also led to instability, likely because the experts conflicted across different tasks. Although $N=1, G=1$ may face similar task conflicts, its larger expert capacity helps mitigate this problem (similar to SDXL). In contrast, with $N=4, G=0.25$, tasks can select non-conflicting experts (see Figure 5), resulting in stable training.
>
> **Why do we need MTU?:**
>  - Our results suggest that multi-task performance is linked to the width (i.e., expert capacity) and depth of the model. Specifically, reducing the width of the experts in shallow models like SDv1.5 works well, however in the case of deeper models like SDXL, larger experts perform well.
> - However, the optimal expert size and count depend on your compute. If you are limited by resources $N=1, G=1$ may provide good results due to large expert capacity; however, if you can accommodate a larger model, an MTU setup with $N=4, G=1$ will lead to even better performance.
> - Importantly, the success of the \(N=1\) setup does not diminish the value of the MoE approach—in fact, our findings show that as the model scale increases, the benefits of MoE become even more pronounced, while still outperforming the alternative of using separate models for all tasks that would require significantly more compute.
>
> We hope this addresses all of the reviewer's concerns, and we thank you for your feedback led to interesting experiments.

---

> > ### Comment · Reviewer_ptLD · 2025-04-03
> >
> > Thanks to the authors for providing a comprehensive rebuttal.
> >
> > I understand that there would be issues in the size of experts in SDXL experiments.
> > To thoroughly validate the effectiveness of experts, can you test the identical experiments with a single expert of large hidden size? (e.g. N=1, G=4)

---

> > > ### Author Response · Authors · 2025-04-07
> > >
> > > Thank you for your comment. Based on your feedback, we experimented with $G > 1$ to increase the size of the experts relative to the pre-trained model. Specifically, we evaluated SDXL using the configurations $N=1, G=4$; $N=1, G=2$; and $N=2, G=2$, where, as in our main training setup, we only fine-tuned the experts, router, and task-specific layers. We present the results in  the Table B below.
> > >
> > > | $N$ | $G$ |      T2I      |      IE      |      SR      |      IP     |
> > > |---|--|--------------|-------------|--------------|------------|
> > > |         |        |FID ($\downarrow$) | I-T Dir Sim ($\uparrow$)|LPIPS ($\downarrow$)| I-I Dir Sim ($\uparrow$)|
> > > | 1 | 1 | 3.9 |20.1 | 26.5 | 44.2|
> > > |1 | 2 | 6.3 | 19.8 | 26.6 | 44.5|
> > > |1 | 4 | 13.7 | 18.9 | 30.2 | 36.7|
> > > |2 | 2 | 5.8 | 20.4 |  24.8 | 44.7|
> > >
> > > Table B: Ablation over the size of experts when $G>1$
> > >
> > >
> > > The $N=1, G=2$ and $N=2, G=2$ configurations reduce performance metrics for T2I while slightly improving the performance of other tasks. However, compared to the results with a reduced $G < 1$ reported in Table 4 of the main paper, increasing $G$ still preserves the performance of many image-to-image tasks, indicating that our model remains scalable for these tasks. Moreover, with $G=2$, increasing the experts from 1 to 2 boosts the performance of all tasks, suggesting the importance of the MTU setup for multi-task learning in diffusion models.
> > >
> > > That said, the MTU model does not perform well with the $N=1, G=4$ configuration for either model, as increasing the model width by four times while keeping the rest of the network frozen appears to negatively impact training stability. We also experimented with training the entire model, but this approach also resulted in unsatisfactory results. We believe this occurs because SDXL is a large pre-trained model trained with a specific FFN size. Modifying the expert size without relatively adjusting the *size* of the rest of the architecture disrupts training stability.
> > >
> > > **Is it better to increase the size of the experts or the number of experts?** From our experiments, increasing the expert size beyond that of the pre-trained model (i.e., setting $G>1$) leads to a decline in T2I performance, while image-to-image tasks maintain performance comparable to $G=1$. Moreover, when $G=1$, increasing the number of experts boosts performance across all tasks (see Table A in our rebuttal). We believe this occurs because SDXL, a large model T2I pre-trained with a specific FFN size, is sensitive to changes in expert dimensions. Modifying the expert size without adjusting the size of the rest of the architecture disrupts training stability and adversely affects the pre-training task (T2I). In contrast, image-to-image tasks—learned from the initialization—benefit from a slightly increased $G$ as it adds more capacity, but suffer when $G$ is increased to 4 as other model parameters are not well-tuned for this expert size. For the same reason, image-image performance may be reduced with the shrinking of the expert sizes due to the overall loss of model capacity. It may be that for very large or smaller expert sizes to be effective, SDXL must first be pre-trained with larger FFNs than those currently used. In contrast, shrinking the expert sizes may have worked for SDv1.5 because using a smaller FFN may also result in an optimal pre-trained model. Alternatively, it could be the case that current SDv1.5 has some redundant FFN parameters, therefore, $G=0.25$ works well. Investigating optimal FFN sizes for pre-training is out of scope of this work. Therefore, we recommend maintaining the expert sizes (i.e., keeping $G=1$) and increasing the number of experts for improved performance across all tasks.

---

### Official Review · Reviewer_zrKU · 2025-03-12

**Overall Recommendation:** 3

**Summary:**

The paper aims to achieve multi-tasking ability for a pre-trained T2I model, via a "lightweight" approach - MTU.
The paper starts from the insight that Feed-Forward Networks (FFNs) receive the most significant change when finetuned to a new task for a pre-trained T2I model.
MTU splits original FFNs into smaller FFN experts and uses a dynamic router mechanism to adapt to new tasks.

## update after rebuttal
My concerns are resolved after the rebuttal. I keep my original rating.

**Claims And Evidence:**

One of the most important conclusions is authors claim that FFNs in the pre-trained T2I models receive the most dramatic change when adapted to downstream tasks. The authors demonstrate this insight by visualizing the deviations from previous weights in Fig. 2, supporting that FFNs are specializing in adapting to new tasks.

**Essential References Not Discussed:**

N/A.

**Experimental Designs Or Analyses:**

The authors compare in different tasks to demonstrate the performance of MTU and its versatility. MTU achieves the best in most of the tasks. Given that MTU is the only approach handling 4 different tasks, it is promising the proposed techniques are generalizable to different downstream tasks.

The authors also show an analysis of expert in Figure 5, which is helpful to understand the mechanism of router assignment.

**Methods And Evaluation Criteria:**

The methods are intuitive based on the insights in Table 1 and Figure 2. Due to the specialization of FFNs, MTU uses them as expert and use a router as a kind of modulation block to better adapt to different new tasks.

For evaluation criteria, authors evaluate three different tasks, using FID, LPIPS, Directional similarity as metrics.  However, for tasks like Super-Resolution, only LPIPS is evaluated while some traditional metrics such as PSNR and SSIM are not considered.

It might be helpful to add a small user study to demonstrate the superiority of MTU.

**Other Comments Or Suggestions:**

- In Fig. 3, it might be helpful to label each component, e.g., "Input image", "Output image", "Encoder", "Decoder". Also might be more clear if authors can exactly show what the prompt is in Fig. 3.
- In Fig. 1, it might be helpful to indicate which T2I model is used here.

**Other Strengths And Weaknesses:**

Weaknesses:
- Somehow all the image results shown in the paper are in a low quality. For example, in Figure 2, both cases show a low quality (e.g., blurry background, over-saturated color style) in the first place. Is it because of the pre-trained T2I models' poor performance? So in Figure 3 (wolf) and IE in Figure 4. This becomes a concern that the improvement brought by MTU may be trivial.

**Questions For Authors:**

Please see my previous sections.

**Relation To Broader Scientific Literature:**

N/A.

**Theoretical Claims:**

No theoretical claims are found in the paper.

---

> ### Author Rebuttal · Authors · 2025-03-31
>
> Thank you for your positive review, and for recognizing that MTU is the only approach so far to handle 4 different downstream tasks. We address your concerns below.
>
> 1. **Quality of image generation:** We believe our MTU model produces images of comparable quality to the base model. While Figure 3 does show some instances of over-saturation in the inpainting (IP) images (left columns, bottom row) for MTU model, the same over-saturation can also be seen in single-task models. The appendix includes several additional  examples where this is not an issue. We point to Figure 7 in the appendix, where we show qualitative results of text-to-image generation for SDv1.5 and SDXL, where the quality of the images matches that of the base model. Images selected in the main paper and the appendix were selected at random from the test set.
>
>
> 2. **Clarifications about Figures 1 and 3:**  In Figure 1, the top row displays results from SDv1.5, while the bottom row features outputs from SDXL. In Figure 3, the image editing prompt used is "turn into a polar bear."
>
> Please let us know if you have any more questions. Thank you.

---

### Official Review · Reviewer_iNBi · 2025-03-14

**Overall Recommendation:** 3

**Summary:**

The paper proposes Multi-Task Upcycling (MTU), an extension of pretrained text-to-image models for multi-task on-device deployment. The authors first investigate the differences between fine-tuned weights and pretrained initial weights across different layers in LDM. Based on this observation, they split the Feed Forward Networks (FFNs) into smaller blocks and add components for task-specific processing. MTU achieves state-of-the-art performance for SD 1.5 and SDXL in downstream image synthesis tasks.

**Claims And Evidence:**

1. The proposed multi-task learning method achieves better performance while preserving TFLOPs and parameters compared to other baselines.

2. Inst-Inpaint can only remove objects, and the authors follow the dataset to train MTU. However, MTU has a lower I-I Directional Similarity than vanilla Inst-Inpaint in SD 1.5. Although the performance of other tasks is improved, inpainting is not. The authors must analyze why this happens.

**Essential References Not Discussed:**

No.

**Experimental Designs Or Analyses:**

The experimental designs are valid.

**Methods And Evaluation Criteria:**

Benchmark and evaluation metrics are reasonable.

**Other Comments Or Suggestions:**

1. It would be better to report the performance of the base model in Tab. 1.

2. Please add qualitative comparisons for text-to-image generation in the main paper. Currently, the text-to-image generation results are only presented quantitatively.

**Other Strengths And Weaknesses:**

**Strengths**

1. The analysis of each block (FFN, CA, SA) in LDMs is interesting.

**Weaknesses**

1. If the base model changes (e.g., LCM, SD3.5, Flux), the weight distribution of FFNs could differ.

**Questions For Authors:**

1. The number of training samples is 118K for T2I, 281K for image editing, 23K for super-resolution, and 90K for image inpainting. Doesn’t this data imbalance affect the training results?

2. How is the training batch composed? Does each training batch consist of different tasks or the same task? And what is its effect?

**Relation To Broader Scientific Literature:**

The proposed model is more applicable to real-world on-device applications since its number of parameters is almost the same as that of the base diffusion models, yet it can handle multiple tasks with a single model.

**Theoretical Claims:**

This paper lacks any proof or theoretical claims.

---

> ### Author Rebuttal · Authors · 2025-03-31
>
> Thank you for your review. We have answered your questions below.
>
> **Weight distributions of FFNs in SD3.5 for image-to-image tasks:** Thank you for your comment. To the best of our knowledge, general image-to-image generation using MMDiT-based models has not been thoroughly explored. Although there is some work in image editing [1], this work was released as open source only after our submission, and its applicability to other image-to-image tasks remains unclear. To address your question, we designed a baseline approach that adds separate positional embeddings to the source and target images, concatenates them along the feature dimension, and processes the extra channels with a linear layer while fine-tuning the entire model. While this baseline produces decent results, we have not included them in our paper, as image-to-image generation in MMDiTs is a research topic in its own right.
>
>  To understand the distribution of weights within our proposed baseline, we fine-tuned SD3.5 on super-resolution (SR) and image editing (IE) tasks, then measured the distance between the fine-tuned and pre-trained models for the FFNs and various attention layers. The table below reports the mode of the distance across all layers, revealing that the FFNs deviate most from the pre- trained initialization. While we see some outlier layers in some attention layers, the deviation is consistently high in FFNs. This suggests that these layers should be split into experts for MTU.
>
>
> |       | FFN   | attn-Q  | attn-K  | attn.-V | attn-Out | attn2-Q | attn2-K | attn2-V | attn2-Out |
> |----|-------|---------|---------|---------|----------|---------|---------|---------|----------|
> | SR | **1.032** | 8.7e-3  | 9.5e-3  | 9.6e-3  | 0.344    | 7.4e-3  | 4.5e-3  | 4.6e-3  | 0.320    |
> | IE  | **0.856** | 6.7e-3  | 7.6e-3  | 5.1e-3  | 0.318    | 0.01    | 8.3e-3  | 4.5e-3  | 0.320    |
>
> Table A: Mode of the deviations between the fine-tuned and pre-trained model for SDv3.5.
>
> **Training data and batches:** Each training batch incorporates all tasks, and we minimize the sum of all losses in each iteration. We opted against random task selection per batch to avoid increased training time. During training, we construct the dataloader using the smallest dataset (super-resolution) and then sample subsets of equal size from the other datasets, shuffling these subsets each epoch. On average, this approach maintains balance across datasets.
>
> **Inpainting performance:** Thank you for pointing this out. Reviewer FqBN asked about our analysis of task interference, and we addressed this issue in our response to Reviewer FqBN. In summary, we analyze model performance by evaluating different combinations of tasks to understand how they interfere with one another. Our findings indicate that inpainting (IP) is less compatible with text-to-image generation (T2I) and super resolution (SR), and it is highly compatible with image editing (IE). This incompatibility may contribute to the observed performance drop in IP. We view reducing task interference as an important direction for future work and will address it in the revised version of the paper.
>
> **Qualitative results for text-to-image generation:** We will add the results for the text-to-image generation task in the main paper. These results are currently presented in the appendix (Figure 7).
>
> [1] FreeFlux: Understanding and Exploiting Layer-Specific Roles in RoPE-Based MMDiT for Versatile Image Editing, Wei et al.

---

> > ### Comment · Reviewer_iNBi · 2025-04-05
> >
> > Thank you for your rebuttal. I have read it carefully, along with the other reviews.
> >
> > Please include detailed training schemes and an analysis of task interference in the revised version.
> >
> > As my concerns have been addressed, I will increase my original rating from 2 to 3.

---

> > > ### Author Response · Authors · 2025-04-07
> > >
> > > Thank you for increasing your score. We will add more details and task interference analysis in the revised version of the paper.

---

### Decision · Program_Chairs · 2025-05-01

**Decision:**

Accept (poster)

**Comment:**

This paper receives positive ratings of (3, 3, 3, 3). The rebuttal has resolved most of the reviewers' concerns, and the AC finds no reason to overturn the decisions of the reviewers. An acceptance is recommended.